# PET/MRI in the Presurgical Evaluation of Patients with Epilepsy: A Concordance Analysis

**DOI:** 10.3390/biomedicines10050949

**Published:** 2022-04-20

**Authors:** Katalin Borbély, Miklós Emri, István Kenessey, Márton Tóth, Júlia Singer, Péter Barsi, Zsolt Vajda, Endre Pál, Zoltán Tóth, Thomas Beyer, Tamás Dóczi, Gábor Bajzik, Dániel Fabó, József Janszky, Zsófia Jordán, Dániel Fajtai, Anna Kelemen, Vera Juhos, Max Wintermark, Ferenc Nagy, Mariann Moizs, Dávid Nagy, János Lückl, Imre Repa

**Affiliations:** 1PET/CT Outpatient Department, National Institute of Oncology, H1122 Budapest, Hungary; 2Medicopus Healthcare Provider and Public Nonprofit Ltd., Somogy County Moritz Kaposi Teaching Hospital, H7400 Kaposvár, Hungary; emri.miklos@med.unideb.hu (M.E.); toth.zoltan@sic.medicopus.hu (Z.T.); daniel.fajtai@sic.medicopus.hu (D.F.); moizs.mariann@kmmk.hu (M.M.); janos.luckl@sic.ke.hu (J.L.); repa.imre@sic.medicopus.hu (I.R.); 3Division of Nuclear Medicine and Translational Imaging, Department of Medical Imaging, Faculty of Medicine, University of Debrecen, H4032 Debrecen, Hungary; 4National Cancer Registry, National Institute of Oncology, H1122 Budapest, Hungary; kenessey.istvan@oncol.hu; 5Department of Pathology, Forensic and Insurance Medicine, Semmelweis University, H1091 Budapest, Hungary; 6Department of Neurology, Medical School, University of Pécs, H7623 Pécs, Hungary; toth.marton@pte.hu (M.T.); janszky.jozsef@pte.hu (J.J.); 7Accelsiors Ltd., H1222 Budapest, Hungary; j.singer@accelsiors.com; 8Neuroradiology Department, Semmelweis University, H1083 Budapest, Hungary; barsi.peter@semmelweis-univ.hu; 9Dr. József Baka Diagnostic, Radiation Oncology, Research and Teaching Center, Somogy County Moritz Kaposi Teaching Hospital, H7400 Kaposvár, Hungary; vajda.zsolt@sic.medicopus.hu (Z.V.); bajzik.gabor@sic.medicopus.hu (G.B.); 10Department of Pathology, Medical School, University of Pécs, H7623 Pécs, Hungary; pal.endre@pte.hu; 11QIMP Team, Center for Medical Physics and Biomedical Engineering, Medical University of Vienna, 1090 Vienna, Austria; thomas.beyer@meduniwien.ac.at; 12Department of Neurosurgery, Medical School, University of Pécs, H7623 Pécs, Hungary; doczi.tamas@pte.hu; 13Department of Neurology and Neurosurgery, National Institute of Mental Health, Neurology and Neurosurgery, H1145 Budapest, Hungary; fabo@mail.oiti.hu (D.F.); jordan.zsofia@mail.oiti.hu (Z.J.); kelemen.anna@oiti.hu (A.K.); nagy.david.gergo@mail.oiti.hu (D.N.); 14Epihope Non-Profit Kft, H1026 Budapest, Hungary; epilepszia.juhos@gmail.com; 15Department of Neuroradiology, MD Anderson, Houston, TX 77030, USA; max.wintermark@gmail.com; 16Department of Neurology, Somogy County Moritz Kaposi Teaching Hospital, H7400 Kaposvár, Hungary; nagy.ferenc2@kmmk.hu

**Keywords:** epilepsy surgery, medically refractory focal epilepsy, presurgical evaluation, MRI-negative patients, discordant electroclinical and MRI data, metabolic PET, hybrid [^18^F]-FDG PET/MRI, preoperative workflow, concordance analysis, epilepsy team

## Abstract

The aim of our prospective study was to evaluate the clinical impact of hybrid [^18^F]-fluorodeoxyglucose positron emission tomography/magnetic resonance imaging ([^18^F]-FDG PET/MRI) on the decision workflow of epileptic patients with discordant electroclinical and MRI data. A novel mathematical model was introduced for a clinical concordance calculation supporting the classification of our patients by subgroups of clinical decisions. Fifty-nine epileptic patients with discordant clinical and diagnostic results or MRI negativity were included in this study. The diagnostic value of the PET/MRI was compared to other modalities of presurgical evaluation (e.g., electroclinical data, PET, and MRI). The results of the population-level statistical analysis of the introduced data fusion technique and concordance analysis demonstrated that this model could be the basis for the development of a more accurate clinical decision support parameter in the future. Therefore, making the establishment of “invasive” (operable and implantable) and “not eligible for any further invasive procedures” groups could be much more exact. Our results confirmed the relevance of PET/MRI with the diagnostic algorithm of presurgical evaluation. The introduction of a concordance analysis could be of high importance in clinical and surgical decision-making in the management of epileptic patients. Our study corroborated previous findings regarding the advantages of hybrid PET/MRI technology over MRI and electroclinical data.

## 1. Introduction

The precise localization of epileptic foci and mapping the relation to the eloquent cortical areas is a prerequisite for the successful presurgical evaluation of patients with pharmacoresistant focal epilepsy [1,2]. Long-term scalp video-electroencephalography (VEEG) monitoring to record ictal EEG and seizure, semiology, neuropsychological assessment, magnetic resonance imaging (MRI), interictal [^18^F]-fluoro-deoxyglucose ([^18^F]-FDG) positron emission tomography (PET) imaging are relevant constituents of this workflow [3,4,5,6]. The epileptic patients with concordant electroclinical data may have a chance at seizure freedom in approximately 30–90% of cases [7,8,9]. In the rest of the patients, MRI findings appeared to be normal or discordant with VEEG and clinical data, and they may benefit from intracranial EEG (icEEG) recordings for the localization of the seizure onset zone [7,8,10]. [^18^F]-FDG PET mapping holds promise for evaluating both temporal [11,12,13,14] and extra-temporal lobe epilepsy [15].

Clinical decision-making is particularly challenging in patients with discordant neuroimaging and electroclinical data, with MRI-negative results, or with the occurrence of multiple epileptic foci. Furthermore, the complexity of electroclinical and neuroimaging data challenges presurgical decision-making [2,6,14,16].

The optimal presurgical diagnostic work-up of epilepsy patients remains a subject of debate, despite significant advances in diagnostic imaging techniques, such as MRI and PET imaging and, distinctively, hybrid PET/MRI [3,4,5,17,18,19,20,21,22,23,24,25,26,27,28].

The aim of our prospective study was to evaluate the clinical impact of hybrid [^18^F]-FDG PET/MRI on the presurgical evaluation of patients with pharmacoresistant epilepsy and to introduce a mathematical model from the multi-modality tests that may facilitate the development of artificial intelligence for the analysis of different concordance patterns.

## 2. Materials and Methods

### 2.1. Subjects

This prospective study was approved by the Scientific Research Ethics Committee of the Medical Research Council (008899/2016/OTIG) and carried out in accordance with the Declaration of Helsinki of the World Medical Association. Seventy patients with refractory focal epilepsy underwent a full electroclinical presurgical evaluation between June 2016 and December 2017. The inclusion criteria were: (i) pharmacoresistant focal epilepsy, (ii) MRI scans with discordant results or without noticeable morphologic epileptogenic lesion, (iii) VEEG monitoring in each patient, and (iv) age of 18–65 years. Exclusion criteria included: (i) standard contraindications for MRI examinations, (ii) acute non-epileptic neurological disorder, (iii) acute infection, and (iv) serious comorbidities. Ten of these patients were excluded from further analysis after the multidisciplinary team revealed multifocal or diffuse pathological alterations (encephalitis *n* = 7, vasculitis *n* = 2, and hydrocephalus *n* = 1). One more patient was removed from the current analysis because of compromised image quality. The median age of the remaining 59 patients was 33 years (range: 18–57 years), and the cohort contained 35 male and 24 female patients.

### 2.2. Patient Preparation

All epileptic patients were hospitalized for adaptation a day prior to the study, and a standard neurological examination, electrocardiography (ECG), and routine laboratory tests were performed. Written consent was obtained from all participants. Dual-modality [^18^F]-FDG PET/MR imaging was performed the next day. The standardized patient preparation for the PET examination was performed according to the European guideline of 2009 [29]. Briefly, supervision of a 2 h duration and VEEG monitoring (in 10–20 EEG Placement) were performed before the intravenous tracer administration. VEEG monitoring covered the whole uptake period of the tracer to ensure the interictal state. PET/MRI acquisition started 60 min after the injection.

### 2.3. PET/MRI Acquisition

All PET/MRI acquisitions were performed on a Biograph mMR scanner (Siemens Healthineers, Erlangen, Germany). The detailed dedicated seizure protocol of MRI acquisition is summarized in Table 1. In order to provide a complete temporally and spatially correlated PET dataset, a 20 min and 35 min list-mode 3D PET acquisition was performed simultaneously for each patient. Vendor-provided UTE sequence was used for PET attenuation correction (AC) purposes, and MR-based attenuation maps were generated automatically. Static image reconstruction was performed both for 20 min and 35 min. AC and non-AC transaxial slices were generated. For PET image reconstruction, the OP-OSEM method was applied, including PSF correction (3 iterations, 21 subsets, 4 mm full-width at half-maximum (FWHM) Gaussian filtering, and 344 × 344 × 127 imaging matrix). µMaps were checked for potential artifacts, and the completed PET raw data were archived for further evaluation. For the current assessment, a 20 min static PET image dataset was used.

### 2.4. Image Processing

An in-house image processing pipeline was applied to transform all individual images into the MNI152 atlas space prior to the regional analysis of the [^18^F]-FDG PET images using Statistical Parametric Mapping (SPM). At the beginning of this procedure, we used the “recon-all” pipeline of FreeSurfer software (version 7.0) for the segmentation of T1-MPRAGE images [30,31,32]. The produced segmented T1-MPRAGE images were used for correcting the misalignment of PET/MR image pairs, global voxel intensity scaling, and calculating the transformations required by the spatial normalization. In the latter case, we applied the FSL software package (version 6.0) [33] and the Advanced Normalization Tools software (version 2.3.5) [34] for calculating the rigid body, 12-parameter affine, and non-linear transformations. After the transformations of the [^18^F]-FDG PET images into the MNI152 space, to eliminate the inter-subject variability of the measured global-brain metabolism according to the standard PET-SPM method, we set the average of the within-brain mask voxel-values of the PET images to 50 [35]. Finally, on the normalized [^18^F]-FDG PET images, we applied a 10 mm and 2 mm 3D Gaussian kernel-based smoothing for the SPM and the regional analysis, respectively.

We used the spatially standardized, globally normalized, and smoothed [^18^F]-FDG PET data and the spatially standardized T1-MPRAGE and T2-FLAIR images for calculating 15 quantitative image-processing parameters for all patients with four image-processing methods (Table 2). The quantitative image-processing parameters were evaluated by VOI (volume of interest) analysis, asymmetry index calculations, SPM analysis, and MAP07 analysis using the spatially standardized, globally normalized, and smoothed [^18^F]-FDG PET, and the spatially standardized T1-MPRAGE and T2-FLAIR images.

During this study, two regional systems in the MNI152 space were applied: the Harvard-Oxford Cortical and Subcortical Atlas (HOVOI), containing 124 (96 cortical and 28 subcortical) regions suitable for regional analysis, and the 14 regions, combined from HOVOI’s regions, used in electroclinical data evaluation (EPIREG system) [36]. All quantitative image-processing parameters were converted into these regions for the purpose of statistical and concordance analysis.

The minimum, maximum, mean, median, and standard deviation (voi.min, voi.max, voi.mean, voi.median, and voi.sd) of the regional [^18^F]-FDG values for all HOVOI regions were estimated in the VOI analysis procedure of the [^18^F]-FDG PET images. The VOI parameters of the overlapping HOVOI regions were used for the regional characterization of the EPIREG system by selecting the minimal value in the case of the voi.min parameter and maximum values in the other cases (Table 2). The maximum values were applied to ensure that the highest average, median, and standard deviation HOVOI data were used to characterize the appropriate EPIREG area, thus preserving the regional variability of the [^18^F]-FDG PET and composite z-score images.

An asymmetry index (AI) calculation of the [^18^F]-FDG PET images was used on symmetric regions of the HOVOI system by applying the formula AI = 100 × 2 × (L − R)/(L + R), where L and R represent the mean intensity values (ai.mean) of the corresponding left and right regions of the HOVOI system. Additionally, using a similar formula, the asymmetry of the maximum, median, and standard deviation (ai.max, ai.median, and ai.sd) were evaluated using a similar formula (Table 2).

An HOVOI-based regional analysis of the Student-t maps was performed by the SPM12 software [37]. A Student-t map was created for each patient using the statistical comparison of their [^18^F]-FDG PET image and the reference metabolic PET image database from our lab, which was built from a previously recorded data pool of 19 cases showing normal PET/MRI patterns. The maximum of the Student-t values and the volume of the hypometabolic region were deployed for characterizing the regional properties of the Student-t maps, sorted by an uncorrected *p* < 0.001 as a threshold (spm.max, spm.vol) (Table 2).

An HOVOI-based regional analysis of the “Composite z-score” images was performed by MAP07. Morphometric analyses were applied to the T1-MPRAGE and T2-FLAIR MRI data sets of the patients using the MAP07 software [38]. The maximum, mean, median, and standard deviation estimates (map.max, map.mean, map.median, and map.sd) were used for characterizing the regional properties of the “Composite z-score” images (Table 2).

The visual analysis of the PET images was performed and analyzed by the authors, KB and ZT, and the MRI images by the authors PB and ZV.

### 2.5. Clinical Data

Electroclinical information and the results of the visual analysis of the PET and MRI images were extracted from patient documentation. Additional PET/MRI investigations were applied for the EPILOBE region-based statistical and concordance analysis (Table 3). According to the possible therapeutic options (resective surgery, neuromodulation, and new antiepileptic drugs), the experts of the epilepsy team (EPI team) categorized the patients by two methods using clinical decision (CD): “Grouping Method 1” (CD1): eligible for resective surgery (without icEEG investigation) and defined as “operable” (7 patients), considered for icEEG exploration and defined as “implantable” (38 patients), or not eligible for any further invasive procedures and defined as “inoperable” (14 patients). During the “Grouping Method 2” (CD2), the simplification of categorization was performed for “inoperable” (14 patients) vs. “eligible for invasive treatment” (45 patients) groups.

### 2.6. Statistical Comparison of Electroclinical and Image Processing Data

Non-normal distribution of the investigated variables was confirmed by the Shapiro–Wilk test. Hence, to assess the relationship between these data and the categorical clinical parameters, non-parametric Mann–Whitney or pairwise Wilcoxon tests were performed. After the statistical analysis, *p*-values were adjusted to control the False Discovery Rate (FDR) [39], and significant relations were selected by the corrected *p* < 0.05 criteria. All statistical analyses were performed by R version 3.6.3. (The R Foundation).

### 2.7. Concordance of the Clinical Data

Using the EPILOBE region-wide electroclinical (Semiology, interictal EEG-iiEEG, and ictal EEG-iEEG) and expert-based imaging data (MRI1, MRI2, and PET.vis), we constructed a localization observation matrix according to the 14 brain regions and the six diagnostic parameters. We excluded the most frequent iiEEG (iiEEG.mfl) and the most frequent iEEG (iEEG.mfl) localization parameters to avoid the over-representation of the ictal and interictal EEG observations. This type of data fusion is suitable for interobserver-analysis regarding different diagnostic procedures, including the independent observations and different regions of EPIREG. Gwet’s AC1 statistics was chosen for the agreement analysis since it was demonstrated to be insensitive to small differences [40,41].

Gwet’s AC1 parameters helped to assess the agreement between different ratings, thus enabling the definition of a new parameter for clinical data concordance (CDC). For our study, the value of the CDC was between 0 and 1, whereby 0 meant “full discordance” and 1 stood for “full concordance.” The performance of the CDC parameters was assessed by means of patient categories, similar to the expert-made clinical decisions-based classification (“eligible for resective surgery,” “considered for icEEG,” “not eligible for any further invasive procedures”). Eight CDC values (electroclinical data (EC), EC + MRI1, EC + MRI2, EC + PET, EC + PET + MRI2, EC + MRI1 + PET + MRI2, EC + MRI1 + MRI2, and EC + MRI1 + PET) were assessed, applying two types of patient classifications (CD1 and CD2).

## 3. Results

### 3.1. Quantitative PET and MRI Analysis

Examples of the results of the presurgical evaluation tests with pathologic findings and the corresponding circular plots of the presurgical data demonstrating different patterns of concordance are shown in Figure 1A–D.

The statistical analysis resulted in 28 significant (FDR-corrected *p* < 0.05) regional associations between the image processing data and clinical data (Table 4). Visual PET investigations (PET.vis) of the regional data correlated with the metabolic-PET asymmetry parameters and the maximal Student-t value of the SPM analysis. The visually localized lesions in the MRI component of the PET/MRI (MRI2) measurements correlated with the PET asymmetry indexes; however, they did not correlate with the MAP07 data. The interictal EEG (iiEEG) localization significantly correlated with the VOI analysis data and the MAP07 regional maximum values, while the iiEEG.mfl localization presented a statistically significant association with the SPM-detected relative volume of hypometabolism. Semiology- or iEEG-based localization did not show any significant association with the image processing data.

The iiEEG and iiEEG.mfl activity localization significantly correlated with the [^18^F]-FDG regional maximum value asymmetry, the [^18^F]-FDG regional mean, median, and standard deviation, the MAP07 generated “composite z-score” maximum, and the SPM- based estimation of the hypometabolic region of the temporal and frontolateral lobes.

We found that the asymmetry score of the regions was highly correlated with the visually identified lesions, mostly in the temporal and the frontal lobes. Despite the low amount of the cardinality of the normative [^18^F]-FDG PET database (*n* = 19), we could demonstrate that the results of SPM analysis, in the cases of temporal lobe hypometabolism, correlated with the visual findings.

### 3.2. Concordance Analysis

The eight concordance parameters in the CD1-type classification statistical analysis by FDR-corrected *p*-values revealed that neither CDC variant could significantly separate the group pairs (Figure 2). However, a tendency was present; in the case of PET-related CDCs, the “inoperable” group showed a borderline significant difference compared to the “operable” or “implantable” groups. In contrast, when the “operable” and “implantable” groups were integrated into the “invasive” group (CD2 classification), only CDC variants containing PET were able to statistically differentiate between the “invasive” and “inoperable” categories (Figure 3A). Figure 3B illustrates the clinical decision differentiation capabilities of the introduced eight CDC parameters by the *p*-values of the Mann–Whitney applied on the CDC-CD2 analysis tests (controlled for FDR).

## 4. Discussion

In our study, the electroclinical data of patients with drug-resistant epilepsy presented a widely discordant pattern. The aim of our prospective study was to test the performance of dual-modality [^18^F]-FDG PET/MRI in patients with pharmacoresistant epilepsy. Using an objective statistical method, we demonstrated that metabolic hybrid PET/MRI technology may significantly contribute to the clinical decision-making in patients with discordant electroclinical and imaging data. The decisions of the EPI team were based on professional knowledge and skills. However, the decision-making was subjective and carried the potential for diagnostic uncertainty among patients with discordant data, which could be even more challenging in the case of MRI-negative patients [2,14]. For this purpose, we introduced a novel concordance analysis method, which demonstrated that PET matrices are of high importance and well-suited to support clinical decisions, especially the matrices including both PET and 3T MRI.

Numerous previous publications suggested the advantages of simultaneous PET/MRI technology over the diagnostic algorithm, with only MRI and electroclinical data [2,11,12,13,15,17,26,27,28,42,43]; however, recent studies have applied a mathematical model to confirm its reliability. Both statistical and concordance analysis highlighted the role of PET imaging for the non-invasive localization of epileptogenic foci, especially in patients with discordant electroclinical data and MRI scans without a definitive epileptogenic lesion or patients with multiple abnormalities.

A concordance analysis demonstrated that PET/MRI examination is able to differentiate between the “invasive” (eligible for invasive treatment) and “inoperable” groups. PET was particularly important in the selection of inoperable patients and confirmed MRI-positive lesions. MRI-assisted PET post-processing techniques (such as the brain atlas-based asymmetry index calculation and SPM analysis) also held additional supportive value for defining clinical decisions. Comparing the visual PET assessment and quantitative PET data, an association between the asymmetry index parameters and visual PET localization proved to be significant, especially for both temporal lobes.

Albeit MR imaging is fundamental in decision-making, it is not sufficient to differentiate between “operable” and “inoperable” patient groups. Additionally, MAP07 measurements did not provide significant conclusions either. The results of our PET/MRI analysis are in line with previously published data in the literature [2,17,19,20,21,22,24,25,27,28,38,42].

Moreover, besides its good feasibility and proper applicability, the hybrid PET/MRI was justified by reductions in radiation exposure, time savings, anesthesia, simplification of study-related organizational and design factors, a range of personalized diagnostic tests, and a range of comorbidity and medication data that may arise [18,23,43,44].

Another non-invasive alternative for localizing epileptogenic foci is simultaneous fMRI and EEG recording. In our study, the positive predictive value of interictal epileptiform discharges, associated with BOLD changes within 2 cm of the epileptogenic zone, was 78%, and the negative predictive value was 81% [45]. Additionally, EEG-fMRI can distinguish between ictal onset-, spread-, and preictal-related BOLD changes [46,47,48]. Besides the single-pass EEG/PET/fMRI [20], the recently reported sub-second analysis method [49] and the topography-related EEG-fMRI [50] may also improve the detection rate of epileptic foci.

In summary, our model confirmed the relevance of simultaneous PET/MRI for epileptic treatment planning. Additionally, the proposed clinical concordance calculation could support the development of a novel artificial intelligence-based decision system in the near future.

## 5. Conclusions

The fully integrated hybrid [^18^F]-FDG PET/MRI has demonstrated a significant impact on the presurgical evaluation workflow of patients with pharmacoresistant epilepsy. The diagnostic algorithm of presurgical evaluation should not miss the comprehensive compliance of PET/MRI, mainly for precarious, subtle lesions or uncertain metabolic patterns. The introduction of a concordance analysis may help the EPI team in clinical decision- making in the future.

## Figures and Tables

**Figure 1 biomedicines-10-00949-f001:**
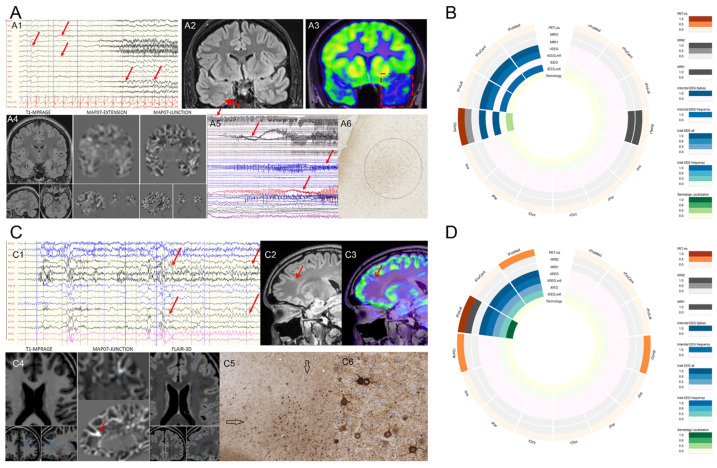
Examples of the results of presurgical evaluation tests proved by pathologic findings. (**A**) A drug-resistant epileptic patient with atypical temporal lobe seizures. (**A1**) Video-EEG monitoring. During her stereotype seizures, left frontotemporal seizure activity was seen (marked with arrows). (**A2**) A cranial MRI showed an FCD2 in the right collateral sulcus (arrow), while (**A3**) [^18^F]-FDG PET/MRI presented a PET hypometabolism in the left temporal lobe (square). (**A4**) The junction map from the MAP07 analysis did not reveal any lesion in the temporal regions. (**A5**) An iEEG monitor was performed because of discordant results. Habitual seizures were registered, and the intervention was conclusive, resulting in a left temporal pole resection (resected region marked with dashed red box) with an Engel I/a outcome (24 months of seizure-free period). (**A6**) Histopathology (NeuN stain) proved an FCD1 in the left temporal pole with irregularly arranged neurons. (**B**) The circular plot refers to the electro-clinical data and imaging modalities of the patient in panel A. (**C**) A drug-resistant epileptic patient with hypermotor seizures. (**C1**) Video-EEG monitoring showed short, stereotype seizures, with left frontal seizure activity (between the arrows). Before the hybrid [^18^F]-FDG PET/MRI study, all MRI investigations were negative. (**C2**) The cranial MRI showed an FCD 2 connected to the left superior frontal sulcus, which was in concordance with (**C3**) [^18^F]-FDG PET/MRI presented a PET hypometabolic pattern. (**C4**) The junction map of MAP07 analysis also detected the lesion (red arrow). Epilepsy surgery with intraoperative electrophysiology was performed targeting this lesion, with an Engel I/a outcome (24 months of follow-up). (**C5**) Histopathology identified an FCD 2a with dysmorphic neurons (arrows; the region is shown in higher magnification in (**C6**) characterized by a lack of anatomical orientation and accumulation of neurofilaments (SMI32, neurofilament immunohistochemistry). (**D**) The circular plot refers to the electro-clinical data and imaging modalities of the patient in panel C. The patterns of presurgical evaluation tests and electroclinical data demonstrated a wide variety of discordances.

**Figure 2 biomedicines-10-00949-f002:**
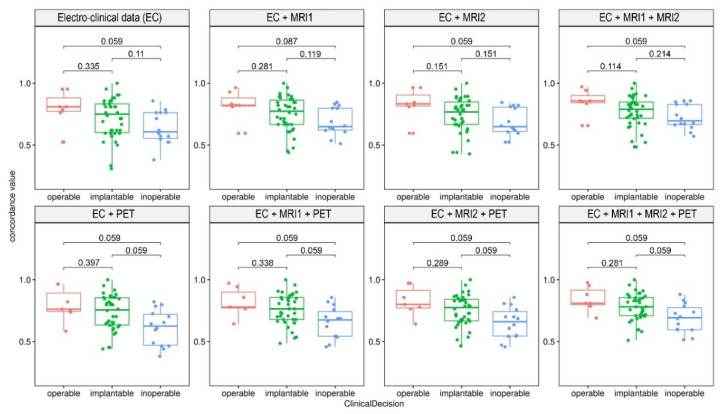
Clinical data concordance (CDC) of “Grouping Method 1.” Boxplots of the eight CDCs grouped by three-way clinical decisions. PET-related measurements showed a slight, but not significant, difference between the “inoperable” versus the “operable” or the “implantable” groups, while PET-independent methods showed relatively less accuracy. Analyzed by Mann–Whitney tests with FDR-correction, *p*-values are shown on the intervals in the charts.

**Figure 3 biomedicines-10-00949-f003:**
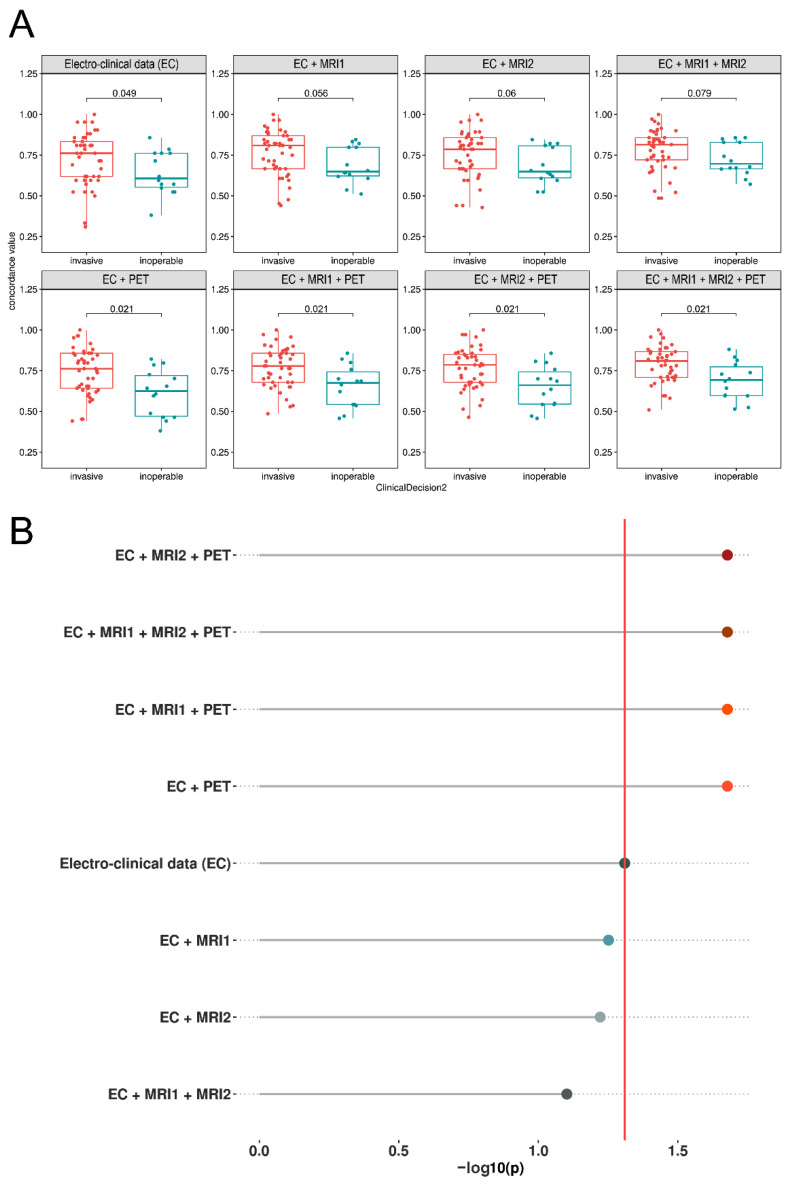
Clinical data concordance (CDC) of “Grouping Method 2.” Boxplots of the eight CDC parameters, depending on the two-way classified clinical decisions (CD2). (**A**) When the “operable” and “implantable” groups were integrated into the “invasive” group, only PET-related CDC variants were able to significantly differentiate between “invasive” and “inoperable” categories. (**B**) Analyzed by Mann–Whitney tests with FDR-correction, *p*-values are shown on the segments in the charts; Negative log10 transformed *p* values also confirmed the high relevance of PET-based measurements since the vertical line corresponding to *p* = 0.05 separates the non-significant and significant comparisons.

**Table 1 biomedicines-10-00949-t001:** Dedicated MRI epilepsy protocol.

MR Sequence	TR (ms)	TE (ms)	FA	Slice Thickness	Imaging Matrix	Voxel Size	TA
Axial T2 UTE (MRAC)	11.94	TE1:0.07, TE:2:2.46	10			1.6 × 1.6 × 1.6 mm	1:38
Sagittal MPRAGE	2300	2.98	9	1.2 mm	240 × 256	1.0 × 1.0 × 1.2	9:14
Axial T2 TSE	6000	106	150	4 mm	358 × 448	0.5 × 0.5 × 4 mm	4:08
Coronal T2 TSE HR	6770	89	150	3 mm	307 × 384	0.5 × 0.5 × 3 mm	3:04
Coronal FLAIR HR	9000	128	120	3 mm	192 × 256	0.9 × 0.9 × 3 mm	5:44
Axial DTI	3600	95	-	4 mm	128 × 128	1.7 × 1.7 × 4 mm	3:59
Axial T2 HEMO	620	19.9	20	4 mm	205 × 256	0.4 × 0.4 × 4 mm	2:09
SagittalT2 SPC 3D	3200	409	120	1.0 mm	261 × 256	0.5 × 0.5 × 1 mm	4:43
Sagittal T2 FLAIR 3D	5000	395	120	1.0 mm	261 × 256	0.5 × 0.5 × 1 mm	5:52
Resting state fMRI	2580	30	90	3 mm	74 × 74	3 × 3 × 3 mm	10:54
GRE Field Mapping	400	4.92/7.38	60	3 mm	64 × 64	3.4 × 3.4 × 3	0:54
Axial ASL	3060.4	17	90	5 mm	64 × 64	3.6 × 3.6 × 5 mm	5:14

**Table 2 biomedicines-10-00949-t002:** Evaluated quantitative [^18^F]-FDG PET image-processing parameters.

Image Processing Data	Description of PET Data	Source
voi.min	minimal [^18^F]-FDG uptake value	the globally normalized and spatially standardized [^18^F]-FDG PET image
voi.max	maximal [^18^F]-FDG uptake value	
voi.mean	average of mean values according to Harvard-Oxford Cortical and Subcortical atlases (HOVOI)	
voi.median	median of HOVOI medians values	
voi.sd	maximal HOVOI based standard deviation	
ai.min	minimum of the asymmetry of minimal HOVOI’s [^18^F]-FDG values	
ai.max	maximum of the asymmetry of maximal HOVOI’s [^18^F]-FDG values	
ai.mean	the maximum value of the asymmetry of HOVOI’s [^18^F]-FDG value means	
ai.median	the maximum value of the asymmetry of HOVOI’s [^18^F]-FDG value medians	
ai.sd	the maximum value of the asymmetry of standard deviations of HOVOI’s [^18^F]-FDG values	
spm.max	highest Student-t value in the HOVOI region	SPM generated Student-t map
spm.vol	the relative volume of hypometabolic area (thresholded by uncorrected *p* < 0.001) in the HOVOI region	
map.max	maximum z-value in the HOVOI region	Combined z-score image produced by MAP07
map.mean	maximum value of the HOVOI’s mean z-values in the HOVOI’s region	

**Table 3 biomedicines-10-00949-t003:** EPILOBE region-wide electroclinical and expert-based imaging data recorded during the study.

Diagnostic Parameters	Description	Value
Semiology	Possible localization considered by semiology in the given EPILOBE region.	0.0: certainly not 0.3: slightly possible 0.6: possible 1.0: the most likely
iiEEG.mfl	Occurrence of interictal EEG activity in the given EPILOBE region (most frequent localization).	0: no 1: yes
iiEEG	Occurrence of interictal EEG activity in the given EPILOBE region.	0: no 1: yes
iEEG.mfl	Possible ictal EEG activity in the given EPILOBE region (most frequent localization).	0.0: certainly not 0.3: slightly possible 0.6: possible 1.0: the most likely
iEEG	Possible ictal EEG activity in the given EPILOBE region.	0.0: certainly not 0.3: slightly possible 0.6: possible 1.0: the most likely
MRI1	Specific epileptogenic MRI lesions found by radiologist experts (before this study).	0: no 1: yes
MRI2	Possible specific epileptogenic MRI lesions found by radiologist experts (during this study).	0.0: certainly not 0.5: possible 1.0: exist
PETvis	Visual PET findings detected by nuclear medicine experts (during this study).	0: no abnormal pattern 0.5: possible 1.0: the most likely

**Table 4 biomedicines-10-00949-t004:** Association between interictal EEG, MRI2, and [^18^F]-FDG PET localization, and [^18^F]-FDG PET image processing data (performed by pairwise Wilcoxon test with FDR adjustment) l: left; r: right; FroMed: frontomedial; FroLat: frontolateral; FroCent: frontocentral; Temp: temporal; Par: parietal; Occ: occipital; Ins: insular.

Source	Image Processing Data	EPILOBE Region	*p*-Value	FDR Adjusted *p*-Value	Meaning in the Detected Lesion
iiEEG	ai.max	lTemp	0.0039	0.0467	lower asymmetry index
	map.max	rTemp	0.0014	0.0172	higher z-score
	voi.mean	rFroLat	0.0020	0.0245	lower [^18^F]-FDG
	voi.median	rFroLat	<0.0001	0.0086	
	voi.sd	rFroLat	<0.0001	0.0025	
iiEEG.mfl	spm.vol	rTemp	0.0040	0.0396	larger SPM hypometabolism area
MRI2	ai.median	rTemp	0.0013	0.0179	lower asymmetry index
	ai.mean	rTemp	0.0016	0.0225	
PET.vis	ai.max	lFroMed	0.0065	0.0276	
		lOcc	0.0166	0.0465	
		lTemp	0.0012	0.0081	
		rIns	0.0076	0.0267	
		rTemp	0.0004	0.0057	
	ai.median	lTemp	<0.0001	0.0004	
		rFroLat	0.0041	0.0145	
		rIns	0.0012	0.0083	
		rTemp	0.0037	0.0145	
	ai.mean	lFroLat	0.0091	0.0254	
		lTemp	0.0002	0.0031	
		rFroLat	0.0067	0.0234	
		rIns	0.0013	0.0060	
		rTemp	0.0006	0.0044	
	ai.sd	lTemp	0.0005	0.0068	
		rFroLat	0.0055	0.0382	
	spm.max	lTemp	<0.0001	0.0012	higher Student-t value
	spm.vol	lTemp	<0.0001	0.0016	larger SPM hypometabolism area
		rTemp	<0.0001	0.0019	

## Data Availability

Database contains personal information. Datasets may be available for special request in anonymized form from Miklós Emri and Imre Repa.

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
