# Peer review of "PET/MRI in the Presurgical Evaluation of Patients with Epilepsy: A Concordance Analysis"

_biomedicines, 2022, doi:10.3390/biomedicines10050949_

Round 1

Reviewer 1 Report

The study performed hybrid PET/MRI on 59 patients with discordant routine presurgical evaluation results. A rating system of clinical data and the PET/MRI was developed and a mathematical model was used to predict the classification of surgical decisions. The authors demonstrated that using this concordance calculation model, including both PET and 3T MRI matrices, can significantly improve the clinical decision.

This is an interesting and methodological sound study. The main concern I have is that despite statistical significance in the invasive versus inoperable model (Fig. 3), there are still many overlap of the concordance values. This may undermine the clinical utility of this model.

The other suggestion is that expert consensus of clinical decision may also be biased, has the authors compared the prediction performance of the model on the outcome of epilepsy surgery?

Nevertheless, this study provides further evidence and directions of incorporate PET information and mathematical model into the decision of epilepsy surgery.

Reviewer 2 Report

The authors aimed to evaluate the clinical impact of hybrid ([18F]-FDG PET/MRI) on the decision workflow of epileptic patients with discordant electroclinical and MRI data. Several issues should be addressed befor publication.

Title is confusing and represent the main issue of the proposed approach. Which is the benefit of PET/MRI acquisition? The article seems to strengthen the added value of PET more than simultaneous PET/MRI in discordant cases.

The abstract is not structured.

- The kind of patients, and so application field, is not specified until the last sentence (epileptic patients).

- Please avoid un-useful capitals "The aim of our prospective study was to evaluate the clinical impact of hybrid [18F]-fluoro- deoxyglucose Positron Emission Tomography/Magnetic Resonance Imaging ([18F]-FDG PET/MRI) on the decision workflow of patients with discordant electroclinical and MRI data".

- Sentences are too much generic and not-informative (i.e., " The concordance analysis provided significant results supporting further therapeutic and surgical decisions": which results?).

Methods - Inclusion criteria are misleading.

- line 97: "One more patient was removed from the current analysis because of technically insufficient PET data". Why if PET is not within inclusion criteria? Why did they include also 1.5T if Biograph PET/MR is a 3T scanner?

-line 111: "The detailed dedicated seizure protocol of MRI acquisition is summarized in Table 1." Table concerns PET post-processing and not MR I acquisition protocol.

-line 112: please clarify this sentence with tool-corresponding scanning time "In order to ensure a complete simultaneous scanning coverage a 20 min and 35 min list- mode PET and MRI acquisition was performed in each patient."

- line 114: ". Vendor-provided T2 UTE sequence was used for PET attenuation". UTE is a T1-weighted sequence.

- line 126: which atlas was chosen for parcellation? Destrieux atlas? HOVOI (please add reference)? How the authors took into account partial volume effects due to this choice ?

- line 162: "Additionally, using a similar formula, the asymmetry of the maximum, the median and the standard deviation (ai.max, ai.median, ai.sd) were evaluated (Table 2)". Should it be referred to table 1?

- How many electrodes for the vEEG acquisition? Experimental setup?

- "Additional PET/MRI investigations were applied for the EPILOBE region-based statistical and concordance analysis (Table 3)". Should it be referred to table 2?

- Which measures were extracted by MRI?

Results: Please consider to rearrange figures to make them more readable (vertical layout for Fig.1 ?).

Discussion: More efforts should be spent to discuss previous studies, also with simultaneous EEG/fMRI tool.

Reviewer 3 Report

The manuscript by Borbély et al. investigated [F-18]FDG PET in the patients with epilepsy.

Major comments

  1. In the “Materials and Methods,” I couldn't understand methodology of [18F]FDG PET at all. The half-life of [18F] is 2 hours. However, the authors described that patients were performed [18F]FDG PET the next day after “using [F-18]FDG”.
  2. The authors described “PET imaging was performed accord- 104 ing to the European guideline of 2009 [25].” This guideline only described transmission scan using [68Ge] and attenuation correction by CT, although the authors used PET/MRI. The authors did not state full width at half maximum of PET, 2D or 3D mode, injection dose of [18F]FDG, times of emission scan, and eye condition during the scan.

Round 2

Reviewer 2 Report

I'm satisfied by authors'reply.